# Task-aware world model learning with meta weighting via bi-level optimization

**Huining Yuan    Hongkun Dou    Xingyu Jiang    Yue Deng***
School of Astronautics, Beihang University, Beijing, China
{hnyuan, douhk, jxy33zrhd, ydeng}@buaa.edu.cn

## Abstract

Aligning the world model with the environment for the agent's specific task is crucial in model-based reinforcement learning. While value-equivalent models may achieve better task awareness than maximum-likelihood models, they sacrifice a large amount of semantic information and face implementation issues. To combine the benefits of both types of models, we propose Task-aware Environment Modeling Pipeline with bi-level Optimization (TEMPO), a bi-level model learning framework that introduces an additional level of optimization on top of a maximum-likelihood model by incorporating a meta weighter network that weights each training sample. The meta weighter in the upper level learns to generate novel sample weights by minimizing a proposed task-aware model loss. The model in the lower level focuses on important samples while maintaining rich semantic information in state representations. We evaluate TEMPO on a variety of continuous and discrete control tasks from the DeepMind Control Suite and Atari video games. Our results demonstrate that TEMPO achieves state-of-the-art performance regarding asymptotic performance, training stability, and convergence speed.

## 1   Introduction

Reinforcement learning (RL) achieves intelligent behavior by optimizing sequential decision-making through a trial-and-error process (Sutton and Barto, 2018). While RL has shown outstanding success in tasks like Go and video games, the enormous quantity of samples required to train such agents poses great limitations on RL's application in real-world scenarios involving human operators, real robots, or computationally expensive simulators (Moerland et al., 2023).

Model-based reinforcement learning (MBRL) aims to enhance the sample efficiency and generalization capability of RL agents through two interleaved stages: *model learning* and *behavior learning*. In the *model learning* stage, an approximate world model of the environment is learned using real environmental samples to provide the agent with the ability to generate simulated experiences or predict the outcome of actions. In the *behavior learning* stage, the agent learns its policy by interacting with the model without having to take actions in the real environment. This paradigm has received a lot of attention, and significant progress has been made in both model learning and behavior learning. (Sutton, 1991; Ha and Schmidhuber, 2018; Janner et al., 2019; Kaiser et al., 2019; Hafner et al., 2019a; Schrittwieser et al., 2020).

One of the most common approaches to building a model is to learn a deep generative model through maximum likelihood estimation (MLE) on environmental trajectories (Buesing et al., 2018; Hafner et al., 2019b,a, 2020; Ozair et al., 2021). Such models can leverage advances in probabilistic modeling, and can easily be combined with state-of-the-art model-free RL agents for task behavior (Ha and Schmidhuber, 2018; Kaiser et al., 2019). However, MLE reconstructs all information from

---

*Corresponding author

37th Conference on Neural Information Processing Systems (NeurIPS 2023).

environmental observations equally, overlooking the information needed for learning specific task behavior. Since the model is only an approximation of the real environment, model errors can create a gap between maximizing model return and maximizing environment return, leading to poorly learned policies (Lambert et al., 2020). Therefore, it is desirable to minimize such a gap by learning a model that can accurately predict those states that have a higher impact on the agent's task policy.

Driven by this goal, value equivalent models are designed to predict future state (or state-action) values rather than the raw observations, such that the model learns to preserve only value-relevant characteristics of the environment (Farahmand et al., 2017; Schrittwieser et al., 2020; Zhang et al., 2020; Grimm et al., 2020, 2021; Antonoglou et al., 2021; Nikishin et al., 2022). Nonetheless, such models often rely on specially-tailored planning algorithms for behavior learning (Schrittwieser et al., 2020), and may face challenges with implementation and optimization (Farahmand et al., 2017), constraining their scalability and robustness across different tasks and learning strategies. Additionally, the substantial amount of semantic information that is discarded during such value-focused learning can be useful for learning an effective policy.

Is there a favorable trade-off between MLE-based models and value equivalent models, where we can enjoy the merits of both worlds? In this work, inspired by recent advances in meta-learning (Nichol et al., 2018; Shu et al., 2019; Jiang et al., 2022), we propose a bi-level framework for model learning, in which we introduce an additional level of optimization on top of an MLE-based model by incorporating a meta weighter network that assigns importance weights to each training sample in the MLE objective function. The meta weighter is then trained to generate novel sample weights by minimizing a proposed task-aware model loss. Under this hierarchical framework, the meta weighter in the upper level of optimization learns to prioritize those samples with a positive impact on closing the task-relevant gap between the environment and model. The model in the lower level is then forced to focus on important samples, while still learning to reconstruct environmental observations and thus, form a state representation with rich semantic information to facilitate policy learning. We name our framework Task-aware Environment Modeling Pipeline with bi-level Optimization (TEMPO).

We build our bi-level framework TEMPO on top of DreamerV2 (Hafner et al., 2020), a powerful MBRL algorithm with a sequential VAE-like model and an actor-critic agent. A simple approximation of the gradient is proposed to update the meta weighter efficiently during implementation. We evaluate the novelty of TEMPO on challenging visual-based RL benchmarks, including both continuous control tasks from the DeepMind Control Suite (Tassa et al., 2018) and discrete control tasks from Atari video games (Bellemare et al., 2013). Results show that our framework achieves state-of-the-art performance compared with model-free RL algorithms, and exceeds the original DreamerV2 in terms of asymptotic performance, training stability, and convergence speed. Furthermore, we perform ablation studies to demonstrate the advantage of our proposed meta-weighting mechanism.

## 2 Task-aware Environment Modeling Pipeline with bi-level Optimization

In this work, we focus on visual-based RL control tasks, which are commonly formulated as partially observable Markov decision processes (POMDPs) with discrete time steps $t \in [1 : T]$, high-dimensional observations $o_{1:T}$ (usually images in visual-based cases), continuous or discrete vector-valued actions $a_{1:T}$, and scalar rewards $r_{1:T}$. The observations and rewards are generated by the black-box environment $o_t, r_t \sim p(o_t, r_t | o_{<t}, a_{<t})$, with actions generated by the agent $a_t \sim p(a_t | o_{\leq t}, a_{<t})$. The goal of RL agents is to maximize the expected sum of rewards $\mathrm{E}_p(\sum_{t=1}^{T} r_t)$.

In this section, we first briefly introduce the world model from DreamerV2 (Hafner et al., 2020) as the foundation of our work. Then, we propose a loss function for evaluating the task awareness of such a MLE-based world model. Finally, we introduce our main contribution, a bi-level model learning framework where a meta-weighting mechanism is proposed to subtly combine the world model with our task-aware model loss.

### 2.1 The world model

We start with the Recurrent State-Space Model (RSSM) proposed in DreamerV2 (Hafner et al., 2020). The RSSM is a probabilistic graphical model similar to a sequential VAE (Kingma and Welling, 2013; Sohn et al., 2015). It conditions on past observations and actions to model the distribution of state transitions that have occurred in the environment.

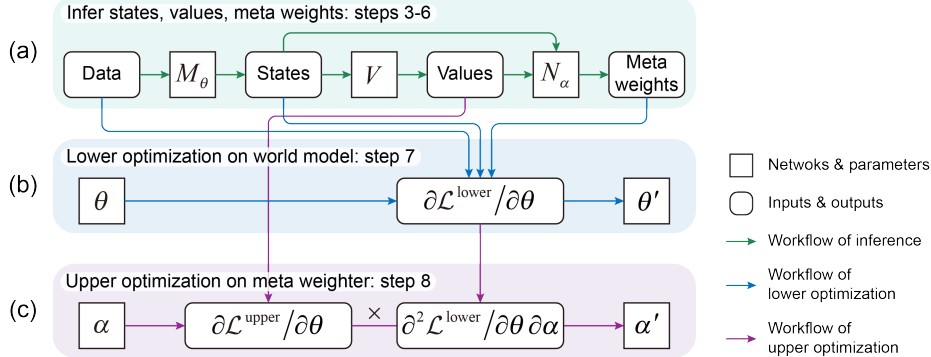

Figure 1: Main workflow of the proposed bi-level model learning paradigm TEMPO, corresponding to steps 3-8 in Algorithm 1. We start with (a) inferring the states and meta weights with model $M_\theta$, value function $V$, and meta weighter $N_\alpha$, then sequentially perform (b) the lower optimization on the model and (c) the upper optimization on the meta weighter.

Consider a trajectory $\tau$ in the environment, which comprises a sequence of observations, actions, and rewards $\tau = \{o_t, a_t, r_t\}_{t=1}^T$. The RSSM introduces deterministic states $h_{1:T}$ and stochastic states $s_{1:T}$ as latent variables for each time step. The observations and rewards are then conditionally generated from these states. Specifically, the RSSM consists of 4 main components

$$
\begin{aligned}
&\text{Deterministic state module:} && h_t = f_\theta(h_{t-1}, s_{t-1}, a_{t-1}) \\
&\text{Stochastic state module:} && s_t \sim p_\theta(s_t|h_t) \\
&\text{Observation \& reward module:} && o_t, r_t \sim p_\theta(o_t, r_t|h_t, s_t) \\
&\text{Representation module:} && s_t \sim q_\theta(s_t|h_t, o_t)
\end{aligned}
\tag{1}
$$

The deterministic state module recurrently outputs $h_t$, whereas the stochastic state module, which corresponds to the prior network in a conditional VAE, predicts the prior $s_t$ at each time step. These two state modules work together to perform the state transitions, while the observation & reward module acts as a decoder to reconstruct $o_t$ and $r_t$ from $h_t$ and $s_t$. To enable end-to-end training through variational inference (Kingma and Welling, 2013), an additional representation module, which serves as an encoder, is used to infer the posterior $s_t$. For clarity, we denote the entire model together as $M_\theta$ with parameters $\theta$.

All components of the model are trained to maximize a variational bound on the trajectory log-likelihood (also known as the Evidence Lower Bound, ELBO) using gradient-based methods (Kingma and Welling, 2013). With Jensen's inequality, the variational bound is written as (see the Appendix for the full derivation)

$$
\log p_\theta(o_{1:T}, r_{1:T}|a_{1:T}) \geq \mathcal{L}^{\text{MLE}}(\tau; \theta) = \sum_{t=1}^T \text{ELBO}_t(o_{\leq t}, r_{\leq t}, a_{<t}; \theta)
$$

$$
= \sum_{t=1}^T \left( \underbrace{\mathrm{E}_{q_\theta(s_t|h_t, o_t)}\Big[ p_\theta(o_t, r_t|s_t, h_t) \Big]}_{\text{reconstruction}} - \underbrace{\mathrm{E}_{q_\theta(s_{t-1}|h_{t-1}, o_{t-1})}\Big[ \text{KL}\big[ q_\theta(s_t|h_t, o_t) \,\|\, p_\theta(s_t|h_t) \big] \Big]}_{\text{regularization}} \right)
\tag{2}
$$

Here, we denote the objective function as $\mathcal{L}^{\text{MLE}}$. For each time step, the variational bound consists of two terms, i.e. the reconstruction accuracy of $o_t$ and $r_t$, and the KL-divergence between the variational posterior and the predictive prior as a regularization. During model learning, the RSSM learns the latent dynamics of the environment from real trajectories by predicting future observations and rewards based on past observations and actions. During behavior learning, the RSSM generates trajectories by unrolling the state vectors using the deterministic and stochastic state modules given actions from an agent, allowing the agent to learn task behavior from simulated experiences in the compact latent state space.

## 2.2 The task-aware model loss

The RSSM is a variational world model trained through MLE, which is designed to reconstruct all information in environmental observations, without taking into account the specific task of the agent. This can lead to a gap between maximizing model return and actually maximizing environment return. To understand this, consider a given state-action pair $(s, a)$ and a typical actor-critic RL agent. If the consequent state predicted by the model is off by $\epsilon$ from the ground truth $s'$, the state value predicted by the critic will also have an error $V(s' + \epsilon) - V(s')$. As the actor's objective is to maximize state values, such errors in the predicted state values can lead to a sub-optimal policy. Therefore, if we want to truly align the model and environment for a specific task, it is intuitive to evaluate a model's performance using some type of value-relevant metric.

Different from MLE, Farahmand et al. (2017) proposed Value-Aware Model Learning (VAML), a loss that evaluates a model's performance by the impact of model errors on the value estimation accuracy. Given an environment transition distribution $p$ and its model approximation $\hat{p}$, a distribution over the state-action space $\mu$, and a value function $V$, the VAML loss is written as

$$\mathcal{L}^{\text{VAML}}(p, \hat{p}; \mu, V) = \int \mu(s, a) \left| \underbrace{\int p(s'|s, a)V(s')\text{d}s'}_{\text{environment value estimate}} - \underbrace{\int \hat{p}(s'|s, a)V(s')\text{d}s'}_{\text{model value estimate}} \right|^2 \text{d}(s, a) \quad (3)$$

The primary concept behind VAML is to penalize a model based on the disparity between the state values of predicted states and those of the ground truth states. However, this loss function has several critical limitations when it comes to implementation on variational world models like the RSSM. In particular, VAML requires a value function that works in a pre-defined state space or the original observation space, while in DreamerV2, the critic from the agent works in the compact latent state space of the RSSM. Secondly, resolving or approximating the expectations/integrals is challenging in a realistic RL environment, where the state space is often vast and continuous. Additionally, VAML compels the model to conserve only the value-relevant characteristics, leading to the rejection of a considerable amount of semantic information contained in the environmental observations.

To construct a task-aware model loss suitable for evaluating the RSSM in the latent state space, we explicitly distinguish prior and posterior states, and replace the environment value estimation with values from the inferred posterior states, model value estimation with values from the predicted prior states, and the expectations with an empirical summation over a given dataset $\tau$, resulting in a parsimonious task-aware loss function, which we name V-VAML (Variational VAML)

$$\mathcal{L}^{\text{V-VAML}}(\tau; V, \theta) = \sum_{t=1}^{T} \left| V(s_t^{\text{post}}) - V(s_t^{\text{prior}}) \right|^2 \quad (4)$$

Here, we introduce superscripts to differ posterior and prior states. Notice the states are outputs of the model, and depend on the model's parameters $\theta$ deterministically through reparameterization, i.e. $s_t^{\text{post}}(o_{\leq t}, a_{<t}; \theta)$ and $s_t^{\text{prior}}(o_{<t}, a_{<t}; \theta)$. We leave out the inputs for simplicity.

The intuition of V-VAML loss is quite similar to the original VAML loss, yet far easier to implement given a dataset. It is also evident that $\mathcal{L}^{\text{V-VAML}}$ has some similarities with the KL regularizer in $\mathcal{L}^{\text{MLE}}$, where a large distance between the posterior and prior is undesirable in both objectives. The difference is that $\mathcal{L}^{\text{V-VAML}}$ evaluates the impact of such distance using the value disparity. One could try somehow replacing the KL regularizer with $\mathcal{L}^{\text{V-VAML}}$ to achieve task-aware model learning. We take a different approach and leave that to future works.

## 2.3 The bi-level framework

We now introduce our proposed meta-weighting mechanism and bi-level framework to subtly fuse the RSSM and our task-aware loss function into a hierarchical optimization paradigm.

Our intention is to attain task awareness in model learning, while still maintaining an MLE foundation to preserve abundant semantic information. Inspired by advances in meta-learning (Nichol et al., 2018; Shu et al., 2019; Jiang et al., 2022), we propose to assign each training sample with different

**Algorithm 1:** Task-aware Environment Modeling Pipeline with bi-level Optimization (TEMPO)

**Input:** Environmental trajectory $\tau$, world model $M_\theta$, meta weighter $N_\alpha$, value function $V$, update steps $K$, learning rate for model $\eta$, learning rate for meta weighter $\lambda$

1 **while** $\theta$ or $\alpha$ *not converged* **do**
2    **for** *update step* $k = 1 \ldots K$ **do**
3      Infer states with world model $h_{1:T}, s_{1:T}^{\text{post}}, s_{1:T}^{\text{prior}} = M_\theta(\tau)$ ;
4      Compute values $\{V(s_t^{\text{post}}), V(s_t^{\text{prior}})\}_{t=1}^{T}$ ;
5      Normalize values $\{V(s_t^{\text{post}}), V(s_t^{\text{prior}})\}_{t=1}^{T} \leftarrow \texttt{Norm}(\{V(s_t^{post}), V(s_t^{prior})\}_{t=1}^{T})$;
6      Compute meta weights $w_{1:T} = \{N_\alpha\big(h_t, s_t^{\text{post}}, s_t^{\text{prior}}, V(s_t^{\text{post}}), V(s_t^{\text{prior}})\big)\}_{t=1}^{T}$ ;
7      Update world model $\theta \leftarrow \theta + \eta \cdot \frac{\partial \mathcal{L}^{\text{lower}}(\tau; w_{1:T}, \theta)}{\partial \theta}$ ;
8      Update meta weighter $\alpha \leftarrow \alpha - \lambda \cdot \frac{\partial \mathcal{L}^{\text{upper}}(\tau; V, \theta)}{\partial \theta} \frac{\partial^2 \mathcal{L}^{\text{lower}}(\tau; w_{1:T}, \theta)}{\partial \theta \, \partial \alpha}$ ;
9    **end**
10 **end**

importance weight in a task-aware fashion. To achieve this, we introduce an additional meta weighter network $N_\alpha$ (with parameters $\alpha$) that outputs a meta weight $w_t$ for each $\text{ELBO}_t(o_{\leq t}, r_{\leq t}, a_{<t}; \theta)$ in $\mathcal{L}^{\text{MLE}}$, based on the current states and values (Figure 1(a))

$$w_t = N_\alpha\big(h_t, s_t^{\text{post}}, s_t^{\text{prior}}, V(s_t^{\text{post}}), V(s_t^{\text{prior}})\big) \tag{5}$$

With these meta weights, the model objective changes from $\mathcal{L}^{\text{MLE}}$ to a weighted sum of $\text{ELBO}_t$s. The meta weights as well as the meta weighter are trained to minimize our task-aware model loss $\mathcal{L}^{\text{V-VAML}}$. This adds another level of optimization on top of the default MLE model learning, forming a bi-level framework with two hierarchical optimizations of orthogonal objectives, which we name Task-aware Environment Modeling Pipeline with bi-level Optimization (TEMPO)

$$\min_\alpha \mathcal{L}^{\text{upper}}(\tau; V, \theta^*) = \mathcal{L}^{\text{V-VAML}}(\tau; V, \theta^*) = \sum_{t=1}^{T} \left| V\big(s_t^{\text{post}}(o_{\leq t}, a_{<t}; \theta^*)\big) - V\big(s_t^{\text{prior}}(o_{<t}, a_{<t}; \theta^*)\big) \right|^2$$

$$\text{s.t.} \ \ \theta^* = \arg\max_\theta \mathcal{L}^{\text{lower}}(\tau; w_{1:T}, \theta) = \arg\max_\theta \sum_{t=1}^{T} w_t \cdot \text{ELBO}_t(o_{\leq t}, r_{\leq t}, a_{<t}; \theta)$$

$$\tag{6}$$

Notice that $\theta^*$ depend on meta weights $w_{1:T}$ and parameter $\alpha$ through the gradients of $\mathcal{L}^{\text{lower}}$. Under this bi-level model learning framework, the meta weighter learns to assign importance weight to training samples regarding their impact on minimizing the task-aware model loss; and the model learns the environment dynamics through reconstruction while focusing on important training samples.

In practice, as the task behavior of the agent gradually improves, the state values naturally rise, which will likely lead to larger differences between the posterior and prior state values. If we naively optimize the meta weighter to minimize $\mathcal{L}^{\text{upper}}$ for the upper-level optimization, the meta weighter may learn to output minimal weight for all training samples to stop the values from rising. Since we are essentially interested in the relative importance of the samples, we perform a normalization with moving mean and variance on the state values $\{V(s_t^{\text{post}}), V(s_t^{\text{prior}})\}_{t=1}^{T}$ before computing $\mathcal{L}^{\text{upper}}$.

To implement our TEMPO framework, we alternate between upper-level and lower-level optimization. That is, we first perform the lower-level optimization on the model parameters $\theta$ through a single gradient update (Figure 1(b))

$$\theta' = \theta + \eta \cdot \nabla_\theta \mathcal{L}^{\text{lower}}(\tau; w_{1:T}, \theta) \tag{7}$$

where $\eta$ denotes the learning rate for the model. Then, we perform the upper-level optimization on the meta weighter parameters $\alpha$, for which we can simply calculate $\mathcal{L}^{\text{upper}}$ using $\theta'$ and take the gradient w.r.t $\alpha$

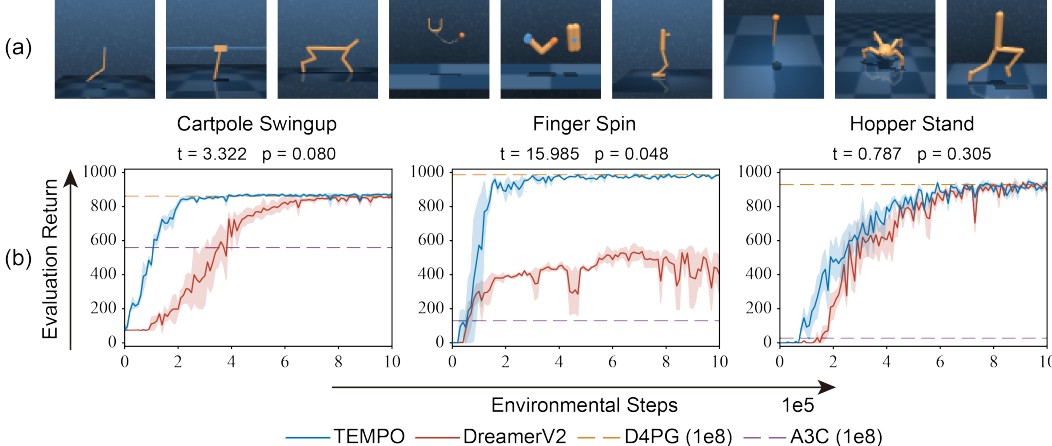

Figure 2: Evaluation of TEMPO on continuous control tasks from DeepMind Control Suite. (a), a graphical demonstration of the 9 environments used in the evaluation, from left to right: Acrobot, Cartpole, Cheetah, Cup, Finger, Hopper, Pendulum, Quadruped, and Walker. (b), the evaluation curves of 3 seeds and their average significance, with lines showing the mean scores, and shaded areas showing the standard deviations.

$$
\begin{aligned}
\nabla_\alpha \mathcal{L}^{\text{upper}}(\tau; V, \theta') &= \frac{\partial \mathcal{L}^{\text{upper}}(\tau; V, \theta')}{\partial \theta'} \frac{\partial \theta'}{\partial \alpha} \\
&= \frac{\partial \mathcal{L}^{\text{upper}}(\tau; V, \theta')}{\partial \theta'} \frac{\partial^2 \mathcal{L}^{\text{lower}}(\tau; w_{1:T}, \theta)}{\partial \theta \, \partial \alpha}
\end{aligned}
\tag{8}
$$

However, the problem with this direct approach is that we need to infer the states (i.e. $h_{1:T}, s_{1:T}^{\text{post}}, s_{1:T}^{\text{prior}}$) twice, one time for each objective, for they come from different model parameters (i.e. $\theta$ and $\theta'$). Therefore, to cut down the computational cost, we propose to empirically approximate $\theta'$ with $\theta$, since they are only one update away. This results in an elegant update formula for $\alpha$ (Figure 1(c))

$$
\begin{aligned}
\alpha' &= \alpha - \lambda \cdot \nabla_\alpha \mathcal{L}^{\text{upper}}(\tau; V, \theta') \\
&\approx \alpha - \lambda \cdot \frac{\partial \mathcal{L}^{\text{upper}}(\tau; V, \theta)}{\partial \theta} \frac{\partial^2 \mathcal{L}^{\text{lower}}(\tau; w_{1:T}, \theta)}{\partial \theta \, \partial \alpha}
\end{aligned}
\tag{9}
$$

It is evident that, with Equation 9, the upper-level objective and lower-level objective are both calculated with the same set of states, which allows us to infer the states only once during an epoch of bi-level optimization. See Algorithm 1 for a pseudocode of our bi-level model-learning paradigm and Figure 1 for a graphical demonstration.

## 3 Experiments

### 3.1 Experimental setup

We implement our TEMPO framework on top of DreamerV2 using the official implementation of Hafner et al. (2020). Specifically, we build the meta weighter to be a 5-layer dense network (MLP) with concatenated states as input (see Equation 5) and scalar meta weight as output. All hidden dimensions of the network are set to 400. Batch normalization (Ioffe and Szegedy, 2015) and ELU (Clevert et al., 2015) activation are performed after each hidden layer. A sigmoid function and an additive bias are applied to the meta weights after the output layer, so that the weights center around 1, i.e. weight $= 0.5 \times \sigma(\text{output}) + 0.75$. An Adam optimizer (Kingma and Ba, 2014) with a learning rate of $1\mathrm{e}{-4}$ is used for updating the meta weighter. We leave the RSSM, the actor-critic agent, as well as all related settings from DreamerV2 untouched and fix the above configuration across all following experiments.

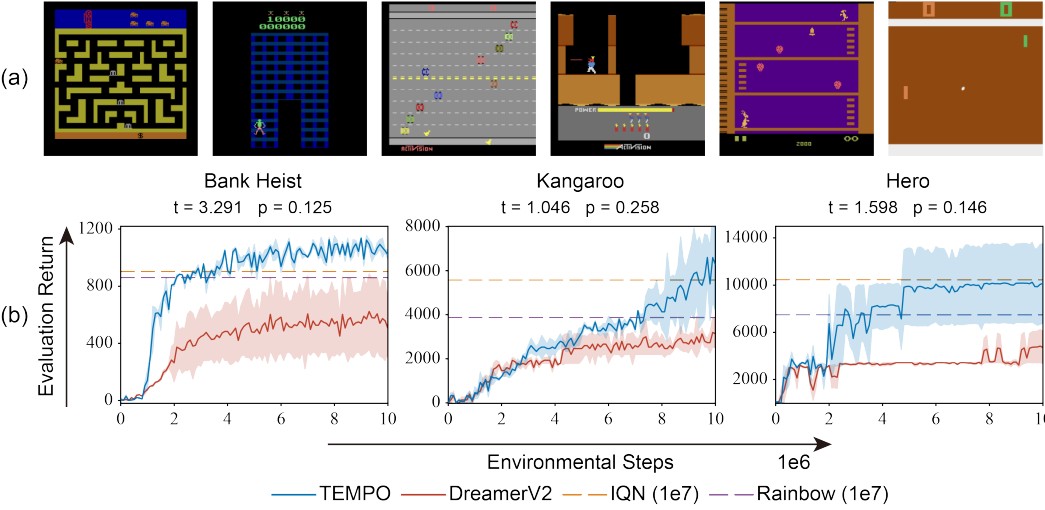

Figure 3: Evaluation of TEMPO on discrete control tasks from Atari video games. (a), a graphical demonstration of the 6 Atari games used in the evaluation, from left to right: Bank Heist, Crazy Climber, Freeway, Hero, Kangaroo, and Pong. (b), the evaluation curves of 3 seeds and their average significance, with lines showing the mean scores, and shaded areas showing the standard deviations.

We use 9 continuous control tasks from the DeepMind Control (DMC) Suite (Tassa et al., 2018) (i.e. Acrobot Swingup, Cartpole Swingup, Cheetah Run, Cup Catch, Finger Spin, Hopper Stand, Pendulum Swingup, Quadruped Walk, and Walker Walk), and 6 discrete control tasks from Atari video games (Bellemare et al., 2013) (i.e. Bank Heist, Crazy Climber, Freeway, Hero, Kangaroo, and Pong), to evaluate the performance of TEMPO.

We pit TEMPO against the original DreamerV2 and a number of state-of-the-art model-free RL agents including D4PG (Barth-Maron et al., 2018), A3C (Mnih et al., 2016), IQN (Dabney et al., 2018), and Rainbow (Hessel et al., 2018). For the model-free agents, we use the results reported by Tassa et al. (2018) and Castro et al. (2018). For DreamerV2, we use the default hyperparameters provided by Hafner et al. (2020) in all experiments. More importantly, we fix these hyperparameters in the corresponding parts of our TEMPO framework (i.e. the RSSM and the actor-critic agent) for a fair comparison.

We tuned TEMPO on DMC Walker Walk and Atari Pong such that meta weighter converges gradually alongside the RSSM. This is to ensure the weighter doesn't reach early convergence on a meaningless state representation. Empirically, TEMPO trains stably, and both objectives reach convergence after sufficient training (Figure 4). Refer to the Appendix for further detailed settings.

Following Henderson et al. (2018), we perform significance testing to examine TEMPO's superiority. In particular, we perform the one-tailed Welch's $t$-test (Welch, 1947) on TEMPO's and DreamerV2's results for every evaluation during the training process, and report the average $t$ and $p$ over all evaluations.

Furthermore, to verify the effectiveness and novelty of the proposed meta-weighting mechanism, we compare TEMPO's meta-weighting mechanism with naively weighting the training samples using the task-aware model losses. We also perform ablation studies on different inputs of the meta weighter network and different hyperparameters.

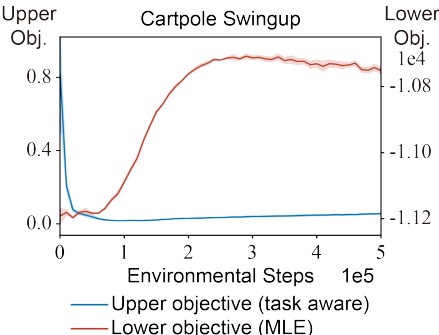

Figure 4: Training curve TEMPO on Cartpole Swingup from DeepMind Control Suite. The lines show the means and shaded areas show the standard deviations of 3 seeds.

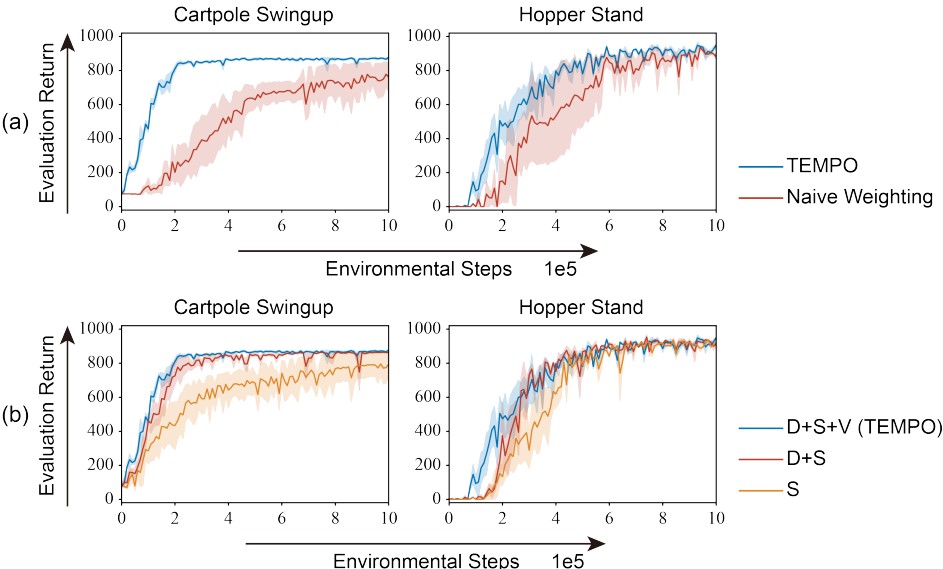

Figure 5: Ablation studies. The lines show the mean scores and shaded areas show the standard deviations of 3 seeds. (a), comparison with naive weighting strategy. (b), Comparison of different meta weighter inputs. Capital letters denote different parts of input: deterministic states (D), stochastic states (S), and state values (V).

## 3.2 Continuous control

A subset of the DeepMind Control Suite with 9 tasks is used to evaluate TEMPO's performance in continuous control situations. The tasks are each from a different environment, as illustrated in Figure 2(a). Environmental observations are RGB images of shape $64 \times 64 \times 3$; actions range from 1 to 12 dimensions; each episode starts with a randomized initial state and lasts for 1000 steps. We follow the protocol described by Hafner et al. (2020) and set the stochastic latent variable (i.e. $s_t$) of DreamerV2 and TEMPO to be a 32-dim continuous vector following Gaussian distribution.

We evaluate the TEMPO agent's environment return every 1e4 environmental steps, and compare the results with those of DreamerV2 and the final performance of D4PG and A3C after 1e8 steps. The results of 3 tasks are illustrated in Figure 2(b), where TEMPO achieved state-of-the-art performance in all tasks. Specifically, TEMPO reached or exceeded the performance of D4PG and A3C within only 1e6 environmental steps, far less than 1e8 steps. Moreover, TEMPO exceeded DreamerV2 in terms of asymptotic performance, training stability, and convergence speed, which demonstrates the advantage of TEMPO's task-aware model learning paradigm under continuous task settings. Refer to the Appendix for full results on all 9 tasks.

## 3.3 Discrete control

We then evaluate TEMPO's performance on discrete control tasks with 6 Atari video games, as illustrated in Figure 3(a). The actions range from 3 to 18 dimensions. We render the environmental observations as gray-scale images of shape $64 \times 64 \times 1$, and set the stochastic latent variable (i.e. $s_t$) to be a 32-column discrete matrix with each column being a 32-dim one-hot vector from a categorical distribution, following Hafner et al. (2020).

Due to limited computational resources, we train TEMPO and DreamerV2 for 1e7 environmental steps. We evaluate the TEMPO agent's environment return every 1e5 steps, and compare the results with those of DreamerV2 and the final performance of IQN and Rainbow after 1e7 steps. The results of 3 games are shown in Figure 3(b), where TEMPO again achieved top performance and exceeded the original DreamerV2. This demonstrates TEMPO's superiority and robustness under both continuous and discrete task settings. Refer to the Appendix for full results on all 6 games.

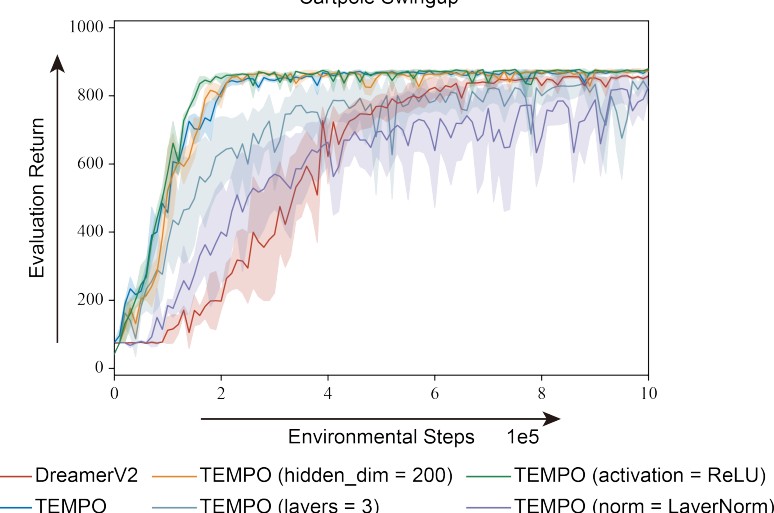

Figure 6: Comparison of different hyperparameters of the meta weighter. The lines show the mean scores and shaded areas show the standard deviations of 3 seeds.

## 3.4 Comparison with naive weighting strategy

In TEMPO, the meta weighter, a deep neural network, generates the meta weights based on state information. Given that the meta weighter is trained to minimize our task-aware model loss, one may naturally wonder: is it viable to naively use the losses themselves as sample weights, assigning greater weights to samples with larger value disparities? To compare this straightforward weighting approach with our TEMPO framework, we initially normalize the task-aware model losses of each time step in a training trajectory using a moving mean and variance. Subsequently, we add 1 to these normalized losses to obtain the final naive sample weights, roughly centering the weights around 1. As illustrated in Figure 5(a), the naive weighting strategy exhibits strong instability during training, and fails to gain comparable results as TEMPO.

## 3.5 Comparison of different meta weighter inputs

In our initial design, the meta weighter generates sample weights by considering three key components of information: deterministic state $h_t$, stochastic states $s_t^{\text{post}}$ and $s_t^{\text{prior}}$, and state values $V(s_t^{\text{post}})$ and $V(s_t^{\text{prior}})$ (see Equation 5). To justify our design, we conduct an ablation study on the inputs to assess the impact of these inputs on the agent's performance. We sequentially remove the state values and deterministic state from the input, and compare the outcomes with our original TEMPO configuration. As illustrated in Figure 5(b), removing the two parts of input causes the agent to converge slower, and introduces additional instability and variance in the agent's performance. Notably, removing the deterministic states severely affects the outcome performance. This demonstrates the significance of the information encapsulated within these deterministic states.

## 3.6 Comparison of different hyperparameters

To examine the robustness of TEMPO's meta-weighting mechanism, we perform simple ablation studies on a number of important hyperparameters of the meta weighter, including hidden dimension, the number of dense layers, the choice of nonlinear activation function, and the normalization method. The results, depicted in Figure 6, clearly demonstrate TEMPO's robustness, especially in the face of reduced hidden dimensions and altered activation functions. While modification to the normalization method and decrease in the number of layers does affect performance, TEMPO can consistently maintain an advantage over DreamerV2 during the initial stages of training, ensuring quicker convergence. Overall, TEMPO proves to be highly adaptable to various hyperparameters.

## 4 Related work

Model learning in partially observable MBRL problems has drawn large interest in related works. A straightforward way for model learning is to learn the dynamics of an environment by fitting the observations from before and after environment transitions, either deterministically or stochastically (Oh et al., 2015; Chiappa et al., 2017; Kaiser et al., 2019). We call this type of model observation-space models. Simulated trajectories can be obtained by unrolling the model in the original observation space, which facilitates behavior learning in a Dyna fashion (Sutton, 1991). For example, SimPLe (Kaiser et al., 2019) proposed a video prediction network for pixel-level prediction on environmental observations, where a PPO (Schulman et al., 2017) agent is trained using the predictions to achieve behavior learning for Atari games.

However, modeling high-dimensional image observations is unavoidably computationally intensive. Latent state-space models overcome this limitation by constructing a compact latent space to characterize the environmental states behind each observation, which minimizes the memory footprint during model unrolling (Watter et al., 2015; Wahlström et al., 2015; Buesing et al., 2018; Gelada et al., 2019; Hafner et al., 2019b,a, 2020; Ozair et al., 2021). Based on the RSSM, Dreamer (Hafner et al., 2019a) and DreamerV2 (Hafner et al., 2020) enable behavior learning by constructing an actor-critic agent that maximizes state values by propagating their analytic gradients back through the dynamics learned by the model. Ozair et al. (2021) proposed a stochastic state-space model based on VQ-VAE (Van Den Oord et al., 2017), which is then combined with Monte Carlo tree search (MCTS (Coulom, 2007)) for task behavior.

Observation-space models and state-space models can both be categorized as MLE-based models that learn environment dynamics through reconstruction. With these models, model errors are likely to compound during unrolling. Value-equivalent models intend to minimize such errors and improve task awareness by focusing on end-to-end value prediction (Tamar et al., 2016; Silver et al., 2017; Farquhar et al., 2017; Farahmand, 2018; Hubert et al., 2021). Specifically, methods like the Predictron (Silver et al., 2017), Value Prediction Network (VPN (Oh et al., 2017)), MuZero (Schrittwieser et al., 2020), and Stochastic MuZero (Antonoglou et al., 2021) aims to form a state representation that can help a parametric value function make accurate value predictions. Other methods like VAML (Farahmand et al., 2017) and VaGraM (Voelcker et al., 2022) design task/value-aware model losses that minimize value prediction disparities from a given value function.

Although we base TEMPO on the RSSM and VAML in this work, TEMPO is orthogonal to the design of MLE-based world models (lower level) and task-aware model loss (upper level). On one hand, one can find a suitable task-aware model loss to evaluate a MLE-based model (e.g. VAML for observation-space models and our V-VAML for state-space models), and build a bi-level optimization to train the model in a TEMPO fashion. On the other hand, one can replace our V-VAML loss with another task-aware model loss (e.g. VaGraM) by making modifications similar to V-VAML.

## 5 Limitation and discussion

We present TEMPO, a task-aware framework for learning world models. The framework is driven by a meta-weighting mechanism and a novel task-aware loss function under a bi-level optimization. TEMPO achieves state-of-the-art performance on a variety of continuous and discrete control tasks, while demonstrating better asymptotic performance, training stability, and convergence speed. Ablation studies are performed to justify our method. The main limitation of our work is that TEMPO is more computationally demanding than previous model learning methods, as it involves two loops of optimization. In our experiments, we observed that TEMPO trains around 40% slower and requires 80% more RAM than DreamerV2. Nonetheless, TEMPO contributes significant insights and opens up an exciting new avenue for environment modeling. We aspire that TEMPO will serve as a catalyst for novel ideas and approaches for model-based RL in the future. A sample code of TEMPO is available at `https://github.com/deng-ai-lab/TEMPO`.

## 6 Acknowledgements

This work was supported in part by the National Natural Science Foundation of China under Grant 62325101 and Grant 62031001.

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

# A Further discussion

## A.1 Task selection

Our primary concern when choosing tasks is mainly the computational cost. With limited computation resources, we couldn't afford to run the extensive 5e6 steps for DMC and 2e8 steps for Atari as the Dreamer papers did. Instead, we only had a 1e6-step budget for DMC and a 1e7-step budget for Atari. Consequently, in the case of DMC, we focus on choosing tasks from different environments on which Dreamer can roughly reach convergence in under 1e6 steps. Similarly, for Atari, we focus on tasks that DreamerV2 could reach convergence within our 1e7 budget. We are certainly enthusiastic about exploring how TEMPO performs on more challenging tasks in the future.

## A.2 Network settings

The architectural configurations of our meta weighter, such as the number of layers and hidden dimensions, were informally aligned with the network modules employed in DreamerV2. Specifically, we mirrored the architectural choices found in modules like the reward module of RSSM, the actor, and the critic, all of which adopt a similar MLP (Multi-Layer Perceptron) structure. Given that these modules can handle state representation well, it was a logical choice to build a similar architecture for our meta weighter as well. Perhaps the most important thing we discovered about architecture design is that batch normalization in the meta weighter greatly boosts the final performance.

For the mapping the network outputs to meta weights, we toyed with $weight = \sigma(output) + 0.5$ and $weight = 0.5 \times \sigma(output) + 0.75$ initial experiments and didn't observe a noticeable difference, so we stuck with the latter in the hope of stable training.

## A.3 Tuning strategy

One of the key pursuits when we built TEMPO was to design a paradigm that could outperform DreamerV2 with little tuning demand. Since TEMPO operates as an additive bonus algorithm, essentially fine-tuning the training sample weights within DreamerV2, it is quite easy for TEMPO to at least meet DreamerV2's performance, resulting in low tuning demand. Specifically, we tuned TEMPO on DMC Walker Walk and Atari Pong such that meta weighter converges gradually alongside the RSSM. This is to ensure the weighter doesn't reach early convergence on a meaningless state representation. We have no doubt that superior architecture and hyperparameters exist for TEMPO, and we remain committed to further exploration and refinement.

## A.4 Performance analysis

Based on our findings, TEMPO demonstrates a notable advantage over DreamerV2 in specific tasks, such as Finger Spin, while in others, like Walker Walk, the two methods exhibit similar performance levels. We hypothesize that the distance between naive state representation from MLE and the task-aware state representation can vary on different tasks. When the distance is large, meaning that the features that can effectively lower reconstruction error don't coincide well with the task-specific features, DreamerV2 can stuck in learning task-irrelevant features like a local minimum, while TEMPO's bi-level learning quickly captures the task-relevant features, and lets the agent performance snowball. In other tasks, where the distance is not significant, TEMPO can have less advantage.

We also observe that TEMPO shows larger variances in some tasks than others. We suspect the reason is that, in these tasks, the meta weighter learns too "eagerly" relative to the RSSM, leading to less meaningful meta weights, potentially disturbing the learning of RSSM. Lowering the learning rate for the weighter may be helpful in reducing the variance in these tasks.

## A.5 Future directions

World models, especially MLE-based models, encode a broad spectrum of environmental dynamics. They carry the potential of multi-tasking with a single model, presenting a possible way to multi-tasking agents, even general AI. With the hierarchical environment modeling paradigm that TEMPO offers, we aspire for it to serve as a foundational framework for enhancing task awareness across multiple specific tasks within future world models.

## B  Derivations

The variational bound, i.e. Evidence Lower Bound (ELBO), for the Recurrent State-space Model (RSSM) with a variational posterior $q(s_{1:T}|o_{1:T}, a_{1:T}) = \prod_{t=1}^{T} q(s_t|h_t, o_t)$, is written as

$$
\begin{aligned}
&\log p(o_{1:T}, r_{1:T}|a_{1:T}) \\
&= \log \int \prod_{t=1}^{T} p(s_t|h_t) \cdot p(o_t, r_t|s_t, h_t) \, \mathrm{d}s_{1:T} \\
&= \log \int \prod_{t=1}^{T} q(s_t|h_t, o_t) \cdot \frac{p(s_t|h_t)\, p(o_t, r_t|s_t, h_t)}{q(s_t|h_t, o_t)} \, \mathrm{d}s_{1:T} \\
&= \log \mathrm{E}_{q(s_{1:T}|o_{1:T}, a_{1:T})}\left[ \prod_{t=1}^{T} \frac{p(s_t|h_t)\, p(o_t, r_t|s_t, h_t)}{q(s_t|h_t, o_t)} \right] \\
&\geq \mathrm{E}_{q(s_{1:T}|o_{1:T}, a_{1:T})}\left[ \sum_{t=1}^{T} \log p(o_t, r_t|s_t, h_t) + \log p(s_t|h_t) - \log q(s_t|h_t, o_t) \right] \\
&= \sum_{t=1}^{T} \left( \mathrm{E}_{q(s_t|h_t, o_t)}\big[p(o_t, r_t|s_t, h_t)\big] - \mathrm{E}_{q(s_{t-1}|h_{t-1}, o_{t-1})}\Big[\mathrm{KL}\big[q(s_t|h_t, o_t) \,\|\, p(s_t|h_t)\big]\Big] \right)
\end{aligned}
\tag{10}
$$

# C Hyperparameters

| Module | Name | Value |
|---|---|---|
| **World model** | Batch size | 16 |
| | Trajectory length | 50 |
| | KL balancing | 0.8 |
| for DMC | Deterministic state dimension | 200 |
| | Stochastic state dimension | 32-dim continuous |
| | Learning rate | 3e−4 |
| | KL scale | 1.0 |
| for Atari | Deterministic state dimension | 600 |
| | Stochastic state dimension | 32-dim discrete (32 classes/dim) |
| | Learning rate | 2e−4 |
| | KL scale | 0.1 |
| **Agent** | Imagination horizon | 15 |
| | Discount factor | 0.99 |
| | Discount factor for $\lambda$-target | 0.95 |
| | Target critic update interval | 100 |
| for DMC | Actor learning rate | 8e−5 |
| | Critic learning rate | 8e−5 |
| for Atari | Actor learning rate | 4e−5 |
| | Critic learning rate | 1e−4 |
| **Meta weighter** | Number of dense layers | 5 |
| | Hidden dimension | 400 |
| | Activation function | ELU |
| | Normalization | Batch |
| | Learning rate | 1e−4 |
| for DMC | Input dimension | $200 + 2 \times 32 + 1 \times 2$ |
| for Atari | Input dimension | $600 + 2 \times 32 \times 32 + 1 \times 2$ |
| **Common** | Optimizer | Adam |
| | Gradient clipping | 100 |
| | Adam epsilon | 1e−5 |
| | Weight decay | 1e−6 |

Table 1: Main hyperparameters of TEMPO. DMC stands for DeepMind Control tasks. We use the default setting of Dreamerv2 in World model, Agent, and Common. All experiments were run on a single Nvidia RTX 3090 GPU with Python 3.7 and Tensorflow 2.6. Refer to our sample code for full implementation details.

## D Full results on continuous control tasks

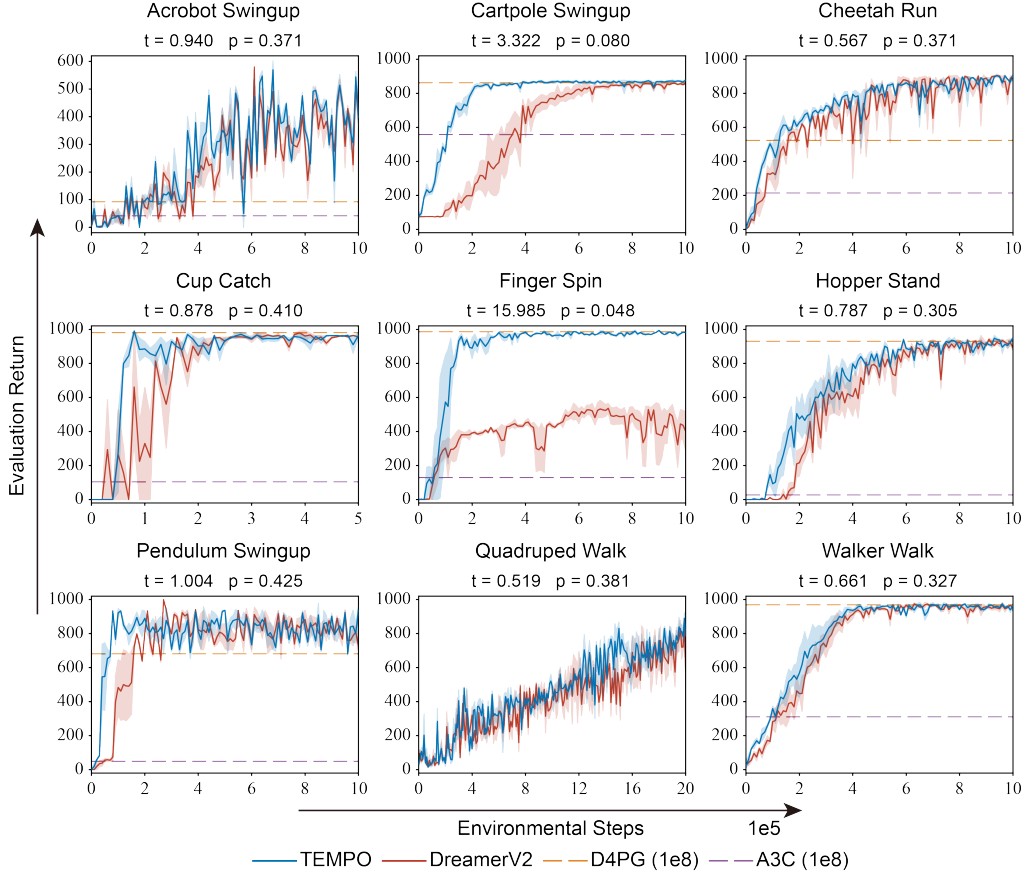

Figure 7: Evaluation of TEMPO on continuous control tasks from DeepMind Control Suite. The lines show mean scores and the shaded areas show the standard deviation across 3 random seeds.

# E Full results on discrete control tasks

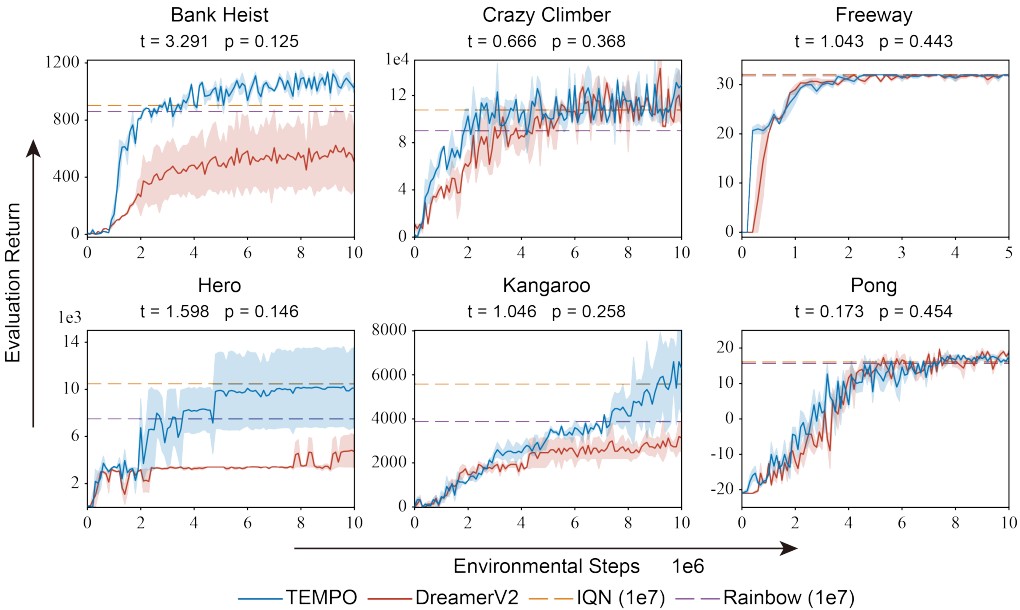

Figure 8: Evaluation of TEMPO on discrete control tasks from Atari video games. The lines show mean scores and the shaded areas show the standard deviation across 3 random seeds.

# F  Training curves

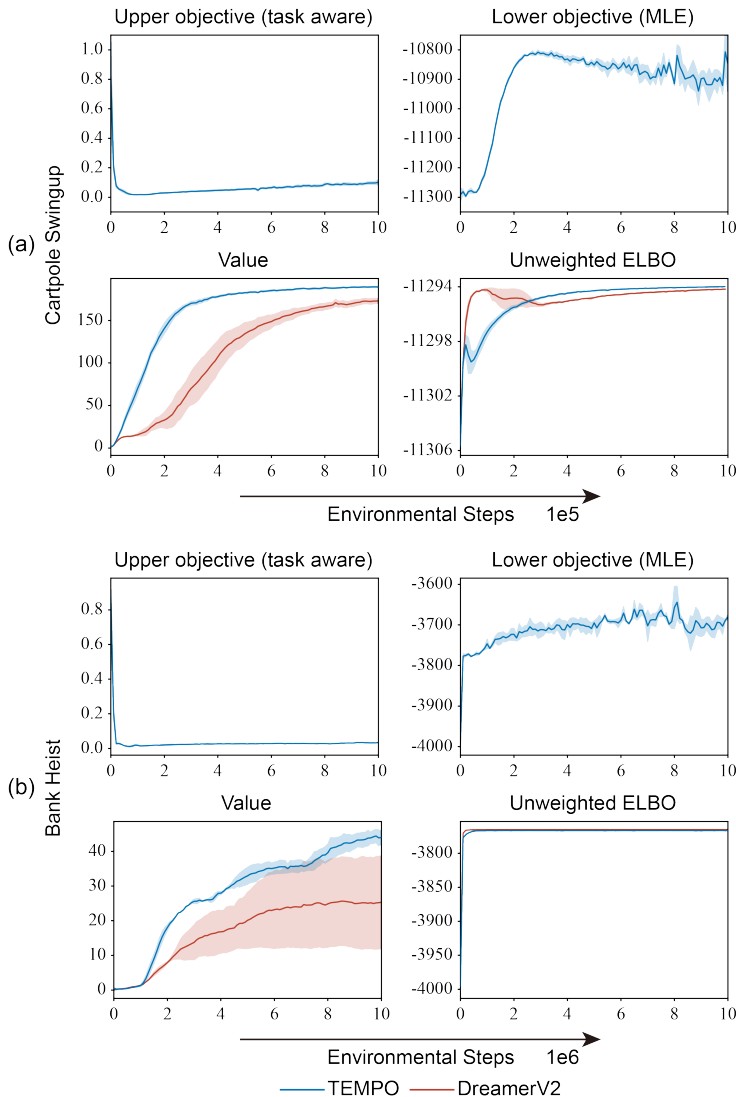

Figure 9: Training curves of TEMPO on (a) Cartpole Swingup from DeepMind Control Suite and (b) Bank Heist from Atari, including curves of value estimation, upper objective, lower objective, and unweighted ELBO. The lines show means and the shaded areas show the standard deviation across 3 random seeds.

## G  Sample weights

| Set | Type | Weights ($w_{1:50}$) | | | | | | | | | Pearson ($r$) |
|-----|------|------|------|------|------|------|------|------|------|-----|---------------|
| **1** | meta | [1.18 | 1.20 | 1.20 | 1.16 | 1.20 | 1.04 | 0.99 | 0.95 | ...] | -0.20 |
| | naive | [1.57 | 0.27 | 0.35 | 1.16 | 0.52 | 1.07 | 0.43 | 0.78 | ...] | |
| **2** | meta | [1.17 | 1.06 | 1.08 | 1.13 | 0.97 | 0.93 | 1.03 | 1.16 | ...] | -0.09 |
| | naive | [0.29 | 1.17 | 0.68 | 0.43 | 0.35 | 1.67 | 1.70 | 0.70 | ...] | |
| **3** | meta | [1.07 | 1.09 | 0.88 | 0.80 | 1.10 | 0.91 | 0.94 | 0.93 | ...] | 0.29 |
| | naive | [2.69 | 4.74 | 1.42 | 1.14 | 2.39 | 0.49 | 0.44 | 0.39 | ...] | |
| **4** | meta | [1.09 | 1.13 | 1.01 | 1.17 | 1.14 | 1.09 | 0.92 | 0.96 | ...] | 0.02 |
| | naive | [0.27 | 0.28 | 0.79 | 0.27 | 0.34 | 0.28 | 0.57 | 0.32 | ...] | |

Table 2: 4 sets of meta weights and corresponding naive weights after $5e5$ steps of training in DMC Cartpole Swingup. The Pearson correlation coefficients show that there is no apparent linear relation between the two types of weights.

