# OpenReview forum: "Task-aware world model learning with meta weighting via bi-level optimization"
_NeurIPS.cc/2023/Conference — NeurIPS 2023 poster_

### Official Review · Reviewer_YURA · 2023-07-04

**Soundness:** 3 good
**Presentation:** 2 fair
**Contribution:** 3 good
**Rating:** 7
**Confidence:** 5

**Summary:**

The authors propose a novel task-aware model learning scheme for model-based RL, which combines the idea of value aware losses with bilevel optimization to obtain a stable training objective from SOTA pixel-based RL approaches such as Dreamer. The method uses the VAML loss as a surrogate objective for learning a weighing function which learns to provide weights to the samples of the ELBO training objective of Dreamer. The loss is therefore able to put more emphasis on samples in which the ELBO needs to be minimized more accurately to prevent large value function differences. The authors evaluate their method with empirical comparisons on some Atari games and DMC environments.

**Strengths:**

The paper proposes a novel approach to task-aware model learning that deals well with previous established problems in the approach and extends well to the more complicated RSSM structures commonly used in pixel-based control.

The paper is overall clearly written without unnecessary notational baggage (see below for improvement suggestions).

The approach seems to offer some improvements in a selection of environments.

The proposed methodology could be extended more, especially with additional experiments clarifying how the learned network improves over the naive weighing strategy, which could make this an important step for future research into the area. This is especially promising since the paper elegantly sidesteps documented problems with the VAML approach.

**Weaknesses:**

The authors (correctly) note that VAML (as is) is inapplicable in a POMDP setting, but propose a method that relies on inferring value functions from the learned "prior" in the RSSM, which by the same logic, should not be doable. Some discussion of this would be good.

Since the paper discusses task aware losses, comparisons to MuZero/EfficientZero (for atari) and TD-MPC (for DMC) would be helpful in understanding the impact of the improvement. This is a non-critical improvement, but it would help contextualizing the results.

The paper does not show whether the loss truly improves (in a VAML sense) over Dreamer. An additional Figure that shows whether the new loss truly results in a lower VAML error and whether this correlates with improved performance would be great.

164: Notationally, the loss $\mathcal{L}-\text{upper}$ is independent of $\alpha$, which is slightly strange. The equations could be rewritten for more clarity, i.e. by moving up equation (8). To make this easier for readers unfamiliar with bilevel optimization, it could also help to reduce the exposition on the RSSM, as most of the details aren't needed for understanding this work and can easily be gathered from the prior work by Hafner, and instead dedicate the space to explaining the bi-level optimization in a bit more detail.

171: It is unclear to me why normalizing the value functions would be sufficient to prevent all weights from becoming small. Are weights also normalized during training? Otherwise wouldn't the algorithm still be incentivized to output 0 weights? Maybe I misunderstood the described problem (as the outer objective does not include the weights, why would it incentivize minimization?).

196: sigmoid-ed -> a sigmoid function is applied.

Figure 2: the chosen tasks are all from the "easy" and "medium" tasks (including those in the appendix) (classification by Yarats DrQ). Why was the suite sub-sampled in this way? Evaluations on hard tasks would be helpful, especially since most environments don't show marked difference between TEMPO and Dreamer. The same question of game selection is also applicable to the Atari benchmark.

Figure 3: why are 6 games shown and only curves for 3?

**Questions:**

For the naive weighing approach: it is unclear from the paper what exactly is gained from the learned weighting over the naive one, especially since the loss seems to be optimized by the naive weight for some mild assumptions. Therefore a qualitative experiment would be very helpful to properly compare: is the learned weight more smooth, are there generalization effects that it is able to take advantage of, etc.?

The authors should discuss the limitations of the proposed approach. Compared to other, more granular approaches such as Vagram (https://openreview.net/forum?id=4-D6CZkRXxI) and TIA (https://arxiv.org/abs/2106.15612), the per-sample weights would not be able to differentiate "RL relevant" dimensions from distracting ones, correct? This should be mentioned and briefly discussed as an important trade-off.

**Limitations:**

The authors describe the limitations shortly with a focus on computational efficiency. For the domain of the paper, this seems sufficient.

---

> ### Author Rebuttal · Authors · 2023-08-04
>
> Dear Reviewer YURA,
>
> We sincerely appreciate your detailed feedback and constructive suggestions! Here is our response to your concerns:
>
> ---
> **Q1: The authors (correctly) note that VAML (as is) is inapplicable in a POMDP setting, but propose a method that relies on inferring value functions from the learned "prior" in the RSSM, which by the same logic, should not be doable.**
>
> **A1:** The main reason for VAML not being straightforwardly applicable in general POMDP settings is that state space is not predefined. One could still use VAML by simply changing the states to observations, and learning the model and value function to work with the original observation space, but doing so is a waste of compact latent space built by RSSM.
>
> The motivation for our modified loss is that, in any VAE-like model, the latent sampled from the prior distribution is considered a predictive latent as the name "prior" indicates, while the latent sampled from the posterior distribution is considered more a "ground-truth" latent as it is inferred from encoded observation. Since the latent space, which is a learned state space, is totally accessible, we propose to modify VAML by replacing the ground truth environmental states with posterior latents, and predicted states from the model with prior latents. The new loss is to evaluate RSSM's performance in the latent space it built, rather than predefined state space like VAML.
>
> ---
> **Q2: Since the paper discusses task aware losses, comparisons to MuZero/EfficientZero (for atari) and TD-MPC (for DMC) would be helpful in understanding the impact of the improvement. This is a non-critical improvement, but it would help contextualizing the results.**
>
> **A2:** More comparisons can definitely be helpful! Please refer to A1 to Reviewer RGcg.
>
> ---
> **Q3: The paper does not show whether the loss truly improves (in a VAML sense) over Dreamer. An additional Figure that shows whether the new loss truly results in a lower VAML error and whether this correlates with improved performance would be great.**
>
> **A3:** As we pointed out in A1, our new loss is essentially VAML loss in the latent space of RSSM. We added a new figure (a modified version of the current appendix E) comparing DreamerV2's training curve with our TEMPO, which can be found in Figure 2 of the rebuttal pdf.
>
> ---
> **Q4: It is unclear to me why normalizing the value functions would be sufficient to prevent all weights from becoming small.**
>
> **A4:** This is a great question! First, a small mistake on our behalf. We weren't being totally informative by writing "The meta weights are sigmoid-ed after the output layer". A bias is also added after the sigmoid so that the weights center around 1, i.e. $\text{weight} = \sigma (\text{output}) + 0.5$ so that the weights range from 0.5 to 1.5, or $\text{weight} = 0.5 \times \sigma (\text{output}) + 0.75$ so that the weights range from 0.75 to 1.25. We tried both in our initial experiments and didn't observe a noticeable difference, so we stuck with the latter in the hope of stable training. We will add this information in the revised version.
>
> Back to the question, without the normalization on the values, the task-aware loss actually rises as training continues, even though the meta weighter is trained to minimize it. This makes sense as the scale of values rises as the agent gets better at the task. However, we observe that most of the meta weights shrink to near the minimal value (i.e. 0.75 in our setting) after some training. We suspect this is because the weighter wants to stop the model from learning environment dynamics, so that the agent can't derive a good task policy from the model, therefore stopping the values from rising. Normalizing the values helps with this problem greatly as it allows the weighter to focus on minimizing the loss within the current value scale.
>
> ---
> **Q5: How were the tasks chosen?**
>
> **A5:** Our concern when choosing tasks is mainly the computational cost. Since we want to run experiments on as many tasks as possible to demonstrate TEMPO's novelty, we couldn't afford to run 5e6 steps for DMC and 2e8 steps for Atari as the Dreamer papers did. We only had a 1e6-step budget for DMC and a 1e7-step budget for Atari given limited time and computation resources. Therefore, for DMC, we focus on choosing tasks from different environments on which Dreamer can reach convergence in under 1e6 steps, and the same thing for Atari with 1e7 steps. We are definitely excited to see how TEMPO does on harder tasks in the future.
>
> ---
> **Q6: Why are 6 games shown and only curves for 3?**
>
> **A6:** The rest are in the appendix due to the space limit. Please refer to A4 to Reviewer kk1X.
>
> ---
> **Q7: For the naive weighing approach: it is unclear from the paper what exactly is gained from the learned weighting over the naive one. A qualitative experiment would be very helpful to properly compare the two.**
>
> **A7:** This is a great question! However, so far, we too are not sure how are the meta weights outperforming the naive weights. We have included a table of a few sets of naive weights and meta weights in Table 1 of the rebuttal pdf. The Pearson correlations between them show that the meta weights are generated in a non-trivial way. We intended to study this question further in the future.
>
> ---
> **Q8: Compared to other, more granular approaches such as Vagram and TIA, the per-sample weights would not be able to differentiate "RL relevant" dimensions from distracting ones, correct? This should be mentioned and briefly discussed as an important trade-off.**
>
> **A8:** Thank you for this great suggestion! VaGraM, as a variant of VAML, does offer the advantage of value-aware model learning in separate state dimensions. It shares the same problem in applicability with VAML. However, similar modifications as what we have done to VAML can be made to VaGraM. The resulting loss can also be used in TEMPO's upper level. We will add more discussion of this in the revised appendix.

---

> > ### Comment · Reviewer_YURA · 2023-08-10
> > **Rebuttal answer**
> >
> > I think I understand your comment about VAML better, I still think it is slightly problematic. The differentiation between state space and observation space is a bit weak in the RL literature, and the way you phrase the statement makes it seem like the partial observability is what the main problem is, not the compressed latent space. Note that you can have compressed encoded latent states without partial observability and vice versa. The "prior" in the RSSM cannot, by construction, encode information from past sequences, it cannot "complete" the partial observable state, but it does make use of the learned encoder. Therefore you are correct that this is preferable, but the way you phrase it in the paper makes it seem like you address a problem resulting from partial observability, which would require you to use a posterior.
> >
> > Nonetheless, I think if you phrase this slightly more carefully, this should not be a problem. I am willing to update my recommendation to accept.

---

> > > ### Author Response · Authors · 2023-08-11
> > > **Response to Reviewer YURA**
> > >
> > > Thank you so much for pointing out our mistake and raising the rating! We will rephrase this problem carefully in the revised version.

---

### Official Review · Reviewer_RGcg · 2023-07-07

**Soundness:** 4 excellent
**Presentation:** 4 excellent
**Contribution:** 3 good
**Rating:** 7
**Confidence:** 4

**Summary:**

The authors present an elegant approach to combine meta-learning and world models. The authors cover the components of their Task-aware Environment Modeling Pipeline with Bi-level Optimization (TEMPO) approach. They then cover experiments across continuous and discrete domains, finding that TEMPO meets or exceeds the performance of DreamerV2. In addition, they present results of ablations to demonstrate the value of their specific approach.

Edit: I appreciate the author's rebuttal and further discussion. I was already positive on the paper and remain so. I did not further increase my rating from an accept as there are issues that cannot be addressed by a revision.

**Strengths:**

This is a very strong paper. It is well-organized, well-written, and well-motivated. The authors identify a timely opportunity to combine meta-learning and world models to gain the values of both. It's perhaps a relatively simple idea, but the execution is at a very high level. The quantity and quality of experiments are also excellent as is the discussion putting those results into context. This is the type of paper I immediately want to share with my students, so it's just a shame I have to wait for it to be accepted!

**Weaknesses:**

The strengths aside, there are a few weaknesses that could be addressed to improve the paper. These keep me from advocating for a higher score than accept (for now). First, while DreamerV2 is a strong baseline, there have been other world model approaches released after it with improved performance. Notably, DreamerV3. The inclusion of these other baselines or at least discussion around these other potential baselines could have benefitted the paper and better justified the choice of DreamerV2. Secondly and relatedly, the related work (Section 4) currently reads more of a list of historical events without any context for how the authors view these works in relation to their own. This is particularly bad at the end, when approaches are just individually described, but no reason given as to why. Finally, there's some interesting results that would have benefitted from more discussion. Specifically, whatever was happening with Hopper Stand. In both Figure 2 and Figure 5 its a clear outlier. Some discussion as to what is happening here would improve the paper, even just in the appendix.

**Questions:**

1. Why did the authors choose DreamerV2 as the one baseline?
2. What do the authors think is happening with Hopper Stand?

**Limitations:**

No negative societal impacts likely, limitations acknowledged.

---

> ### Author Rebuttal · Authors · 2023-08-03
>
> Dear Reviewer RGcg,
>
> We sincerely thank you for giving us encouraging comments and recognizing the contribution of our work. Here are our answers to your questions:
>
> ---
> **Q1: While DreamerV2 is a strong baseline, there have been other world model approaches released after it with improved performance. Notably, DreamerV3. The inclusion of these other baselines or at least discussion around these other potential baselines could have benefitted the paper and better justified the choice of DreamerV2. Why did the authors choose DreamerV2 as the one baseline?**
>
> **A1:** The Dreamer series is one of the most, if not the most, recognized model-based RL algorithms in recent years. Its biggest strength, as we see it, is its generality. It provides state-of-the-art performance in a variety of envs and tasks right out of the box, while being intuitive in design. Perhaps the only other method that can rival the fame and generality of Dreamer is MuZero, although, to our knowledge, no one has pitted them together in benchmarks like DMC and Atari under identical env settings. For TEMPO, we want to inherit the generality of Dreamer and further boost its performance by incorporating value awareness. Given that we were in a time rush for this work, we thought the easiest way to effectively support our method is to set Dreamer as the main baseline. However, we do believe that comparing TEMPO with other methods like MuZero will provide valuable insights into TEMPO, as well as Dreamer.
>
> As for why build TEMPO on top of DreamerV2 rather than DreamerV3, the reason is quite practical. When we found out about DreamerV3, although it has a significant improvement over DreamerV2 in terms of generality, the official implementation is written in Google JAX, rather than PyTorch or TensorFlow, which we have experience with. By that time, we have already done part of our work based on DreamerV2. With limited time, we decided to stick with v2. Then again, for Dreamer, TEMPO is an additive bonus boost algorithm that optimizes the data usage, it is absolutely compatible with v3. We are excited to see TEMPO improve DreamerV3 in the future!
>
> ---
> **Q2: The related work (Section 4) currently reads more of a list of historical events without any context for how the authors view these works in relation to their own. This is particularly bad at the end, when approaches are just individually described, but no reason given as to why.**
>
> **A2:** This is a great suggestion! We will try to polish the related work section and further clarify their relationship to our work in the revised version if we can manage to fit the page limit. For now, readers can still get a general and intuitive idea of the relationship between our work and others in the introduction section.
>
> ---
> **Q3: There's some interesting results that would have benefitted from more discussion. Specifically, whatever was happening with Hopper Stand. Some discussion as to what is happening here would improve the paper, even just in the appendix. What do the authors think is happening with Hopper Stand?**
>
> **A3:** Again, we thank you for this great suggestion! We will add further discussion of our results to the appendix in the revised version.
>
> As we pointed out in A6 to Reviewer 9zSq, we believe the distance between naive state representation from MLE and the task-aware state representation can vary on different tasks. When the distance is large, meaning that the features that can effectively lower reconstruction error don't coincide well with the task-specific features, TEMPO can have a significant advantage (e.g. Finger Spin). We think Hopper Stand is one of those tasks where such distance is small in the first place, and the RSSM itself can capture the task-specific features well enough, so TEMPO shows less advantage.

---

> > ### Comment · Reviewer_RGcg · 2023-08-11
> > **Re: Rebuttal by Authors**
> >
> > I'd like to thank the authors for their comprehensive response.
> >
> > * That justification totally makes sense. Practical issues come for us all! But some discussion in the paper or appendices about this decision and some estimation from the authors about what the difference in results might look like with other approaches would benefit the submission.
> > * I agree that a general idea is possible to read from the current section, and I appreciate the commitment to improve it.
> > * I appreciate the insight as to what's happening! I'd definitely encourage the authors to add this discussion to the appendices.

---

> > > ### Author Response · Authors · 2023-08-11
> > > **Response to Reviewer RGcg**
> > >
> > > We sincerely thank you for the suggestions! We will definitely add more disscussion to the appendix in the revised version. They will surely provide the readers with more insight into our work.

---

### Official Review · Reviewer_kk1X · 2023-07-09

**Soundness:** 2 fair
**Presentation:** 2 fair
**Contribution:** 2 fair
**Rating:** 5
**Confidence:** 4

**Summary:**

This paper proposes a novel model-based algorithm termed Task-aware Environment Modeling Pipeline with Bi-level Optimization (TEMPO). To achieve a proper trade-off between MLE and Value-Equivalence models, TEMPO utilizes a bi-level optimization framework that trains an upper-level weight network generating non-uniform sample weights concerning the value equivalence and a low-level RSSM learning the state representations. The performance of TEMPO is examined in benchmarks involving continuous control tasks and video games.

**Strengths:**

- The paper is well-organized and the proposed method is easy to understand.

- The experiments are conducted in more than one particular benchmark.

**Weaknesses:**

One concern is that the proposed method is not well-motivated, and there are some factual errors in the texts about motivation. As claimed in lines 38-42 and 115-11, the MLE-learned representation overlooks the task-specific information. Actually, the learned representations of Dreamerv2 not only rely on the reconstruction loss but the reward prediction and transition prediction that indeed provides task-specific information. Nevertheless, the claim made in lines 130-131 is not necessarily a limitation of VAML, since we can simply use some existing representation learning technique to convert the obs in POMDP to a low-dimensional state, in which case VAML loss is still applicable.

The experiments with only 3 random seeds are not convincing to me. Only 3/9 results of DMC and 3/6 results of Atari are demonstrated in the main paper, which is not enough. Even though the results in the remaining environments are not promising, the main paper should demonstrate most of them.

Taking the significant computational demand of TEMPO (lines 307-308) into account, I think the demonstrated performance improvement is marginal.

**Questions:**

N/A

---

> ### Author Rebuttal · Authors · 2023-08-03
>
> Dear Reviewer kk1X,
>
> Thank you for your detailed and constructive feedback. Here is our response to your concerns:
>
> ---
> **Q1: The learned representations of DreamerV2 not only rely on the reconstruction loss but the reward prediction and transition prediction that indeed provides task-specific information.**
>
> **A1:** In common RL envs, state transition prediction only provides general information about the fundamental dynamics and physics of those envs. Reward prediction does offer task-specific information. However, the action of agents, either actor-critic agents or value-based agents, are more directly influenced by the value function, which RSSM is unaware of. I suppose the phrase "completely overlooking" is a bit too strong. We will modify this in the revised version.
>
> ---
> **Q2: The claim made in lines 130-131 is not necessarily a limitation of VAML, since we can simply use some existing representation learning technique to convert the obs in POMDP to a low-dimensional state, in which case VAML loss is still applicable.**
>
> **A2:** Yes, but doing so adds additional difficulty. Therefore, not being applicable to common situations right out of the box is a valid limitation.
>
> ---
> **Q3: The experiments with only 3 random seeds are not convincing to me.**
>
> **A3:** We are well aware that using 3 random seeds is not ideal when a lot of other RL works use 5. The reason for this is simply limited time and computation resources. However, TEMPO does outperform DreamerV2 by a great margin in our experiments, which is enough to effectively support our method.
>
> ---
> **Q4: Only 3/9 results of DMC and 3/6 results of Atari are demonstrated in the main paper, which is not enough. Even though the results in the remaining environments are not promising, the main paper should demonstrate most of them.**
>
> **A4:** In fact, TEMPO outperforms DreamerV2 in most of our experiments and meets DreamerV2 in the rest, despite the margin being relatively small in a few tasks. So I'm not sure that "the remaining environments are not promising" does justice to our work. The reason not to demonstrate all curves is simply the 9-page space limit.
>
> ---
> **Q5: Taking the significant computational demand of TEMPO (lines 307-308) into account, I think the demonstrated performance improvement is marginal.**
>
> **A5:** The code is much faster now! Please refer to A7 to Reviewer 9zSq.

---

> ### Comment · Area_Chair_KC3z · 2023-08-16
> **Are you satisfied by the answers?**
>
> Dear reviewer,
>
> Would you please indicate whether the authors' response is satisfactory for you? If not, please engage with the authors, so we can get a better assessment of this work.
>
> Thank you,
> Area Chair

---

> > ### Comment · Area_Chair_KC3z · 2023-08-18
> >
> > I am following up on this. Would you please read the authors' response and indicate whether you agree with them or not?
> >
> > Thank you,
> > Area Chair

---

> > > ### Comment · Reviewer_kk1X · 2023-08-19
> > > **Response**
> > >
> > > The authors address some concerns. Nevertheless, I think the experimental results are not very promising to me, and maybe a more complex mechanism is required to handle the ELBO rather than a linear-weight combination. Overall, I increased the score to 5.

---

> > > > ### Author Response · Authors · 2023-08-19
> > > > **Response to Reviewer kk1X**
> > > >
> > > > Thank you for the reply and raise in score!

---

### Official Review · Reviewer_9zSq · 2023-07-10

**Soundness:** 3 good
**Presentation:** 3 good
**Contribution:** 3 good
**Rating:** 7
**Confidence:** 3

**Summary:**

The authors propose to combine the benefits of value-based model-based methods (like MuZero) with reconstruction-based model-based methods (like Dreamer). They do so through TEMPO, which is a bi-level model-learning algorithm that includes a meta weighter network that weights each training sample with the goal of minimizing a task-aware loss. They present experimental results to support the claims of state-of-the-art performance.

**Strengths:**

1. The proposed idea of combining the best of both worlds with respect to model-based methods is interesting and timely. The idea of using the upper level to weight samples is also of interest.

2. I like that the authors are clearly quite familiar with the related literature. I appreciate the effort and detail provided to summarize the most relevant work in a way that could help even a newcomer to the field understand the current state of the art.

3. In general, I think this paper is well-written and clear. Similar to my comment in strength 2, I think the authors do a nice job of explaining their method in an accessible way.

4. I appreciate that the authors included a concrete discussion section that explicitly mentions the weaknesses of TEMPO.

5. The authors show promising empirical results: the underlying Dreamer-v2 model for TEMPO does not need any additional hyperparameter tuning to enable TEMPO to achieve performance that is at least as good as the vanilla Dreamer-v2 algorithm.

**Weaknesses:**

1. I encourage the authors to make more clear what is their contribution and what is contributed by previous work in Section 2. It is well-written in general, but including things like the details about deriving the variational bound for ELBO may be unnecessary?

2. I would like to see proper significance testing to support the claims of superiority in the paper [Henderson et al., 2018]. I also would like to see a mention of how the presented seeds were chosen for the environments. I would also like to see how the hyperparameters for TEMPO were chosen. I see that the RSSM + actor-critic were untouched, but what about the other architectural and optimization choices (e.g., Adam learning rate)? How were they selected?

[Henderson et al., 2018] Henderson, Peter, et al. "Deep reinforcement learning that matters." Proceedings of the AAAI conference on artificial intelligence. Vol. 32. No. 1. 2018.

**Questions:**

1. Could you not just use a smaller representational capacity since the task-awareness should bias your network toward preserving the most task-relevant information? Or am I missing something?

2. Why do you think TEMPO exhibits more training variance on only some environments (e.g., Figure 2)? And why do you think that some environments yield similar performance between Dreamerv2 and TEMP (e.g., Hopper Stand) while others have a large discrepancy (e.g., Finger Spin)?

3. Given that TEMPO is more computationally demanding, are there any optimization tricks that you believe could help?

4. Is the reason for performing this optimization in a bi-level manner to not force the model to trade-off between preserving value-relevant information and performing reconstruction? The trade-off between reconstructing the different components of the MDP seems to be a major bottleneck in Dreamer, as one must preselect a network capacity that can preserve all components in the hopes that one will capture the value-relevant ones.

**Limitations:**

The paper includes a nice discussion section that includes the main limitation of the work. Although it would be nice to see a more detailed discussion of limitations, I recognize that authors are provided limited space to include all contributions, so I treat the inclusion of any limitation as a bonus toward the paper.

---

> ### Author Rebuttal · Authors · 2023-08-02
>
> Dear Reviewer 9zSq,
>
> We sincerely appreciate your positive comments and feedback, as well as the suggestions for improvement. In response to your questions, we have prepared the answers below:
>
> ---
> **Q1: I encourage the authors to make more clear what is their contribution and what is contributed by previous work in Section 2. It is well-written in general, but including things like the details about deriving the variational bound for ELBO may be unnecessary?**
>
> **A1:** This is a valid concern. The RSSM world model in Section 2.1 is the work of Dreamer. I suppose this can lead to some confusion about our contribution. We will try to highlight our contribution more in the revised version. On the other hand, you could say it is unnecessary to include the derivation of ELBO for RSSM, but we just want to make the paper as self-contained as possible to reduce readers' burden of referring to other works in order to understand ours.
>
> ---
> **Q2: I would like to see proper significance testing to support the claims of superiority in the paper [Henderson et al., 2018].**
>
> **A2:** Following [Henderson et al., 2018], we further conducted a number of significance testing experiments regarding different hyperparameters for the TEMPO meta weighter, including hidden dimension size, number of layers, activation function, and normalization. Other settings remain the same to maintain a fair comparison with the original DreamerV2. The curves are presented in Figure 1 of the rebuttal pdf.
>
> Although showing sensitivity to the number of layers and type of normalization, TEMPO is quite robust overall.  In fact, one of the key pursuits when we built TEMPO is to design a paradigm that can outperform DreamerV2 with little tuning demand. With TEMPO being essentially an additive bonus algorithm that weights DreamerV2's training sample, it is quite easy for TEMPO to at least meet DreamerV2's performance, leading to low tuning demand.
>
> ---
> **Q3: I also would like to see a mention of how the presented seeds were chosen for the environments.**
>
> **A3:** The 3 random seeds were simply fixed to be 0, 1, and 2 in all of our experiments.
>
> ---
> **Q4: I would also like to see how the hyperparameters for TEMPO were chosen. I see that the RSSM + actor-critic were untouched, but what about the other architectural and optimization choices (e.g., Adam learning rate)? How were they selected?**
>
> **A4:** As we mentioned in A2, we built TEMPO to have low tuning demand. Architectural settings like the number of layers and hidden dimensions were quite casually set to be the same with network modules from DreamerV2 like the reward module of RSSM, actor, and critic (these modules all have similar MLP architecture). Since the network architecture in these modules can handle state representation well,  we can naturally use it for our meta weighter as well. Perhaps the most important thing we discovered about architecture design is that batch normalization in the meta weighter greatly boosts the final performance.
>
> As for the learning rate, we tuned it on DMC Walker Walk and Atari Pong such that meta weighter converges gradually alongside the RSSM. This is to ensure the weighter doesn't reach early convergence on a meaningless state representation. We have no doubt that better architecture and hyperparameters exist for TEMPO.
>
> ---
> **Q5: Could you not just use a smaller representational capacity since the task-awareness should bias your network toward preserving the most task-relevant information? Or am I missing something?**
>
> **A5:** Intuitively, yes. We wanted TEMPO as a performance boost to DreamerV2, so we try not to (or rather didn't have the time) poke around with the DreamerV2 part of our algorithm. After all, DreamerV2 has a lot of settings and hyperparameters. We were worried that reducing the model size would result in the need the re-tune other hyperparameters (e.g. learning rate) as well.
>
> ---
> **Q6: Why do you think TEMPO exhibits more training variance on only some environments (e.g., Figure 2)? And why do you think that some environments yield similar performance between DreamerV2 and TEMP (e.g., Hopper Stand) while others have a large discrepancy (e.g., Finger Spin)?**
>
> **A6:** This is a great question. We suspect the reason that TEMPO shows larger variances in some tasks is because, in these envs, the meta weighter learns too "eagerly" relative to the RSSM, leading to less meaningful weights, potentially disturbing the learning of RSSM. Lowering the learning rate for the weighter may be helpful in reducing the variance in these tasks.
>
> We believe that in some tasks, like Finger Spin, there is a significant distance between naive state representation from MLE and the task-aware state representation, and DreamerV2 can stuck in learning task-irrelevant features like a local minimum, while TEMPO's bi-level learning quickly captures the task-relevant features, and lets the agent performance snowball. In other tasks, where the distance is not significant, TEMPO can have less advantage.
>
> ---
> **Q7: Given that TEMPO is more computationally demanding, are there any optimization tricks that you believe could help?**
>
> **A7:** Since the initial submission, we have optimized our code, resulting in a much faster implementation of TEMPO. The new code, while still consuming around 80\% more RAM than DreamerV2, runs only less than 40\% slower than DreamerV2. We will update this number in the revised version.
>
> ---
> **Q8: Is the reason for performing this optimization in a bi-level manner to not force the model to trade-off between preserving value-relevant information and performing reconstruction?**
>
> **A8:** Yes, this trade-off is exactly our motivation. An MLE-based model like RSSM dedicates all of its network capacity to reconstructing all information equally. With TEMPO, We want to hold out some of the network capacity and dedicate them to focusing on reconstructing task-relevant information. So, yes, it is an intended trade-off.

---

> > ### Comment · Reviewer_9zSq · 2023-08-16
> >
> > Thank you to the authors for providing such a detailed response. I wasn't sure where to find the rebuttal pdf, so that limits my ability to gauge the mentioned tradeoffs. Please feel free to point me toward the pdf so that I can better respond.
> >
> > My main remaining concern is reproducibility due to the small number of runs and the choice of seeds. That said, I want to acknowledge that a small number of runs is still meaningful.
> >
> > I would encourage the authors to make the changes promised in the response to myself and other reviewers, such as clarifying which contributions are theirs and which are RSSM/Dreamer's.
> >
> > I would like to keep my score unchanged, as I think this is good paper that would be beneficial to the NeurIPS community.

---

> > > ### Author Response · Authors · 2023-08-16
> > > **Response to Reviewer 9zSq**
> > >
> > > Thank you for your reply! The one-page rebuttal pdf can be found under the *Author Rebuttal by Authors* at the top of the page, which reads as follow:
> > >
> > > ***
> > > **Author Rebuttal by Authors**
> > >
> > > Dear Reviewers,
> > >
> > > Thank you for your comments and suggestions. We hope our response is sufficient to address your questions.
> > >
> > > Please kindly let us know if you have any additional concerns.
> > >
> > > Paper 3253 Authors
> > >
> > > PDF:  pdf
> > > ***

---

> ### Comment · Area_Chair_KC3z · 2023-08-16
> **Are you satisfied by the answers?**
>
> Dear reviewer,
>
> Would you please indicate whether the authors' response is satisfactory for you? If not, please engage with the authors, so we can get a better assessment of this work.
>
> Thank you,
> Area Chair

---

### Author Rebuttal · Authors · 2023-08-05

Dear Reviewers,

Thank you for your comments and suggestions. We hope our response is sufficient to address your questions.

Please kindly let us know if you have any additional concerns.

Paper 3253 Authors

---

### Decision · Program_Chairs · 2023-09-21

**Decision:**

Accept (poster)

**Comment:**

The paper proposes a task/value-aware model-based RL algorithm based on a bi-level optimization. It uses a Dreamer-like architecture, which solves an MLE criteria for model learning, as the low-level optimizer. In order to make this value-aware, it suggests using a VAML-like objective to train a network that weighs each term of the low-level optimization problem.

The consensus among reviewers is that the paper should be accepted. After reading the paper, I also agree. This is a good addition to the task/value/decision-aware model learning literature.

Reviewers provided several comments in how this paper can be improved. Specifically, I'd like to emphasize the followings:

- The number of seeds is 3, which is small. This is especially important for some results where there is a large overlap between figures, for example, Figure 5 in the main body, and Figures 1 and 2 in the appendix. I ask the authors to increase the seed or perform statistical significant tests.

- The contribution of this work and difference with prior work should be made more clear. The related work section is too compressed. I suggest having an expanded section in an appendix that helps position the paper properly.

- It might be worth trying different networks with varying level of capacity. Value-awareness shows its impact especially when the model is not capable to capture all aspects of the model. Given that TEMPO is only using a weighted MLE to tune the model, studying the effect of model capacity can be illuminating. (This is optional.)

- In my own reading of the paper, I feel initially confused by the use of s^post and s^prior in Eq. 4. This is defined in the paragraph after Eq. 1, but as far as I can see, they have not been explicitly defined. Clarification around this and the intuitive difference between these two states, would be helpful.